# Perceived Accessibility, Satisfaction with Daily Travel, and Life Satisfaction among the Elderly

**DOI:** 10.3390/ijerph16224498

**Published:** 2019-11-14

**Authors:** Katrin Lättman, Lars E. Olsson, Margareta Friman, Satoshi Fujii

**Affiliations:** 1CTF Service Research Center, and Department of Social and Psychological Studies, Karlstad University, SE-65188 Karlstad, Sweden; katrin.lattman@kau.se (K.L.); lars.e.olsson@kau.se (L.E.O.); 2Kyoto University, Katsura, Nishikyo-ku, Kyoto 615-8540, Japan; fujii@trans.kuciv.kyoto-u.ac.jp

**Keywords:** elderly, daily travel, perceived accessibility, life satisfaction, PLS-SEM, satisfaction with travel

## Abstract

People are living longer than they did previously, and the proportion of older people is increasing worldwide. This rapid development will have implications for the transport system, in general, and for travel behavior and accessibility to daily activities, in particular. In recent years, both research and politics have drawn the attention of the public to issues affecting the opportunities of the elderly to participate in everyday life. The debate has so far mostly focused on health issues, with limited work having been done on the ability of the elderly to live the lives they want to considering how they travel. With this view, a theoretical model, grounded in a model of travel and subjective wellbeing was developed to explore the role of perceived accessibility in satisfaction with travel and life satisfaction. Empirical data were collected from a sample of 2950 respondents (aged 60–92) from five cities in Northern Europe (Stockholm, Helsinki, Oslo, Copenhagen, Bergen) and analyzed using partial least square structural equation modeling (PLS-SEM). The findings confirmed the link between perceived accessibility, travel satisfaction, and life satisfaction. The findings also showed the role of sociodemographic and travel attributes in perceived accessibility and satisfaction with travel, as well as the moderating effects of different age groups. We conclude that this moderating role played by age clearly indicates that we should not treat the elderly as a homogenous group in research and transport planning.

## 1. Introduction

Today, people are living longer than they did previously, and the proportion of older people is increasing worldwide. In 1950, 384 million people were over the age of 60, corresponding to 8.6% of the population (according to [1]). This figure has since risen to nearly 900 million people (12%). By 2050, older people are expected to account for 22% of the world’s population. This rapid growth will have implications for the transport system, in general, and for travel behavior and accessibility to daily activities, in particular.

This study focuses on the elderly (65 or older) and their general life satisfaction. The main question is whether the perceived opportunities of the elderly to live the lives they want are related to their travel satisfaction and general life satisfaction. During recent years, both research and politics have drawn public attention to issues affecting opportunities of the elderly to participate in everyday life. The debate has so far mostly focused on health issues, with limited work having been done on the ability of the elderly to live the lives they want to considering how they travel, which may include how easy it is to do daily activities, whether or not it is possible for them to do all their preferred activities, and whether or not access to preferred activities is satisfying, taking travel mode use into account. In this study, we relate these issues (termed perceived accessibility) to general life satisfaction in order to better understand the links between perceived accessibility, satisfaction with daily travel, and life satisfaction among the elderly.

People are getting older but also healthier; thus, definitions of “older” or “elderly needs” are not as homogeneous as they were 100 years ago. Given this, understanding and paying attention to different living circumstances of the elderly, their perceived opportunities to live the lives they want to, and their life satisfaction are necessary in transport planning and in the development of transport services suited to the elderly with different needs and preferences. We argue that a confirmed relationship between perceived accessibility in daily travel, satisfaction with travel, and life satisfaction justifies measures aimed at supporting older people to live satisfactory lives, regardless of how they travel.

In Western society the opportunity or willingness to be employed generally decreases after 64, with work no longer being a daily activity for many old people. Instead, the freed-up working time can be used for a variety of activities, such as shopping, training, religious activities, health/self-care, and social activities with friends and relatives. Some researchers claim that we live in a “hypermobile” society that requires a high degree of mobility in order to be able to take part in family/social activities, service, and business [2]. Accessibility and mobility provide opportunities for social interaction [3], which in turn helps counteracting loneliness, isolation, and exclusion by promoting meaningfulness in life and a sense of inclusion (e.g., [4,5]). Previous research on adults shows that participation in daily activities is one of the most important factors for people’s life satisfaction [6]. However, knowledge of the perceived ability of the elderly to participate in their preferred activities remains limited. Furthermore, among the elderly, there is also a knowledge gap concerning the relationship between perceived accessibility, travel satisfaction, and general life satisfaction.

By analyzing data from a European public transport barometer (BEST—Benchmarking in European Service of Public Transport), we have studied the daily travel of the elderly of five different Northern European cities, as well as the relationship between perceived accessibility, satisfaction with travel, and life satisfaction. This research contributes to existing knowledge in several ways. In exploring the links between perceptions of accessibility, satisfaction with travel, and general life satisfaction, we can expect to gain increased insight into the importance of daily travel among elderly travelers. By dividing the elderly into age groups, we also expect to gain increased insight into the importance of these concepts to people undergoing different stages of life, something which may be useful in identifying priority areas when planning an inclusive transport system that promotes wellbeing.

The remainder of the paper is structured as follows: In the second section, the daily travel of the elderly is discussed, highlighting differences in travel behavior during different age stages. This is followed by a section on the theoretical framework. In the fourth section, the data collection and methodology are described. The final section contains the discussion and some recommendations for future work.

### 1.1. Daily Travel

Most studies focusing on the elderly and their daily travel have been carried out in the US [7]. In the US, as in other western countries, car use, number of trips, and the proportion of driver’s license holders among older people have all increased [8]. The proportion of women with driver’s licenses is less than that of men, although the number of women with driving licenses is increasing [9]. The elderly have been identified as a heterogeneous group, with different mobility and travel characteristics [10,11]. Nevertheless, knowledge of the similarities and differences in travel behavior between different segments of the elderly is lacking. Researchers have argued that the “new older generation”, who are today in their 60s, are likely to adopt different travel patterns than their parents [10].

The elderly in western society are car drivers; however, due to increasing age and reduced abilities, their willingness to drive and the number of car trips is tending to decrease [12]. A trend is visible in the increasing age at which people decide to reduce their car driving. Cui, Loo, and Lin [13] present a comprehensive review of previous studies (during the 2000s) around the world (e.g., Denmark, the Netherlands, Canada, Australia, the UK, and the US), which shows how the number of trips and their length decrease with increasing age (often with a significant decrease after 75). The dominant mode of transport among the elderly is the car, which accounts for 84%–91% of all trips. However, there are variations within the group (defined in the vast majority of international studies as over 65) where the car dominates in the 75–84 bracket and primarily among men. As far as type of travel is concerned, it is the social journeys that stand out. Surprisingly, perhaps, the distances of these trips do not decrease significantly with increased age. A unique study conducted in Denver [14] analyzed the travel habits of the elderly in 2009–2010. The findings showed that “shopping trips” and “general errands” increase with age, while work-related trips fall rapidly in relation to the retirement age. Interestingly, social activities (i.e., “visiting friends/relatives” and “civic/religious”) are a less common reason to travel in the oldest age group (85+).

People born after the Second World War (during the so-called “baby boom” between 1946 and 1965) are more car dependent, traveling more and further than previous generations. Cui, Loo, and Lin [13] conclude that the car is likely to continue to be the dominant mode of transport for older people. The car dependency of the elderly will further increase as these post-war children become older. As firmly pointed out by Hensher [15], “The important role that a current driver’s license plays cannot be overlooked in the preservation of mobility and accessibility, …”. Giuliano et al. [16] acknowledged that the elderly are less inclined to use public transport than other age brackets. Currie and Delbosc [17] discuss the role of public transport among the elderly and conclude that neighborhood density and mixed use residential and commercial opportunities are influential factors. Low public transport use has also been explained by functional limitations [18,19]. Non-motorized travel among the elderly is limited for the same reasons (e.g., [20]). A very low cycling share (6%) is noted in North America, where older adults rarely cycle [21], while the situation is slightly more positive in Europe. For example, in the Netherlands, 25% of older adults’ trips are by bicycle [22]. Böcker et al. [23] found that elderly women were more likely to walk and cycle than elderly men.

In summary, older people prefer to drive; the trend here is that they are continuing to do so at ever-increasing ages. Public transport is not an option for the vast majority of old people, slightly more old people choose to go by active modes (walking and bicycling). How perceived accessibility affects older people’s travel satisfaction, and possibly their life satisfaction, is discussed in the following section, which focuses on the theoretical framework for this study.

### 1.2. Theoretical Framework

Satisfaction with daily travel refers to the subjective evaluation of travel, which is affected by both cognitive evaluations (e.g., the evaluation of perceived and physical quality aspects) and affective evaluation dimensions (e.g., emotions triggered by travel) [24]. In theory, travel satisfaction is related to people’s subjective wellbeing, which includes an evaluation of life as a whole (e.g., general life satisfaction [25], see also [26]). For the elderly, this link is supported by a recent thematic synthesis of qualitative research on older people (60+) conducted in the UK between 1998 and 2017 [27]. This synthesis of 12 studies showed that travel indeed contributes to old people’s wellbeing. It has also been shown that access to local facilities and services, including transport, and engaging in hobbies and leisure activities, as well as keeping up with social activities and retaining a role in society, are all closely linked to old people’s wellbeing [9,28,29,30]. The perceived possibilities of accessing activities that are of importance to everyday life has not, however, been clarified thus far in any theoretical frameworks regarding travel and life satisfaction. Perceived accessibility is defined as perceived possibilities of accessing activities that are of importance to everyday life, or in terms of “how easy it is to live a satisfactory life considering how people travel” [31].

When it comes to empirical research on accessibility and life satisfaction of the elderly, there are very few studies focusing on the perceived possibilities of participating in or reaching important activities. Banister and Bowling [32], as well as Nordbakke and Schwanen [33], found that transport opportunities facilitating participation in social activities contribute to the quality of life of the elderly. Nordbakke and Schwanen further conclude, in another study [34], that the consequences of low-quality high-cost travel, or unmet mobility needs, were decreased life satisfaction of the older elderly. Enam et al. [35] discuss how the relationship between life satisfaction and activity participation may be twofold, and that the dimensions may be constantly influencing each other. A study of public transport shows that a lack of possibilities of using public transport for daily travel (due to health issues or service supply) negatively affects “the capability of elderly to carry out everyday activities of value” [36]. A study of bus travelers showed that the elderly (around the age of 68), and people in their thirties, experienced significantly lower levels of perceived accessibility when traveling by bus than did other age groups [31]. Perceived accessibility, in the study referred to, was found to be influenced by frequency of travel, quality, and feeling safe. A study by van der Vlugt, Curl, and Wittowsky [37], of perceived accessibility in Nottingham, UK, found that perceptions of accessibility decreased with age. In the study, participants rated different aspects of their daily accessibility (ease of getting to places, range of local facilities, and accessibility to necessary activities). In contrast to age, which was negatively associated with accessibility, a positive attitude toward public transport was found to be associated with higher perceived accessibility. Moreover, when investigating perceptions of walking accessibility of the elderly and families, in another case in Germany, van der Vlugt et al. [37] found that sociodemographic factors had less explanatory value regarding perceived accessibility than did attitudes. Of the sociodemographic factors, only income and disability had any effect (i.e., not age, gender, migration, car use, or education), while atmosphere, satisfaction with the availability of destinations, and different barriers were all linked to perceived accessibility.

This study contributes to previous research by clarifying the role of perceived accessibility in travel satisfaction and general life satisfaction. The model of Ettema et al. [25], of travel and subjective wellbeing, is a valuable theoretical basis that essentially postulates that satisfaction with travel influences—either positively or negatively—people’s life satisfaction. However, previous research on travel and old people indicates that the ability of the elderly to live the lives they want to considering how they travel (i.e., perceived accessibility) should also be taken into account [32,33]. Thus, a link between perceived accessibility and general life satisfaction was included in the theoretical model (Figure 1). We additionally hypothesize that perceived accessibility influences travel satisfaction in that a higher degree of perceived accessibility (ease of travel, possibilities to travel, and access to preferred activities) increases satisfaction with travel while a lower degree of perceived accessibility decreases satisfaction with travel. Thus, different levels of perceived accessibility are assumed to be of significance to both travel satisfaction and general life satisfaction (see the theoretical model in Figure 1). In their review, Cui et al. [13] show how travel behavior varies during old age, which justifies the hypothesis that age group moderates the relationships between (I) perceived accessibility and satisfaction with travel, (II) satisfaction with travel and life satisfaction, and (III) perceived accessibility and life satisfaction.

To the model, we also add travel attributes (mode use) as these have been indicated to vary in old age. However, very limited research has thus far gone into the role of sociodemographic attributes and travel attributes in perceived accessibility and satisfaction with travel among the elderly, with the findings so far not giving any conclusive answers. Thus, we do not assume that there exists a relationship, adding instead potential paths (indicated by dotted arrows in the theoretical model in Figure 1) between the sociodemographic (gender) and travel attributes (frequency of mode use).

### 1.3. The Present Study

The aim of the present study is to better understand the links between perceived accessibility, satisfaction with daily travel, and life satisfaction by taking the diversity of older people into account. As a basis for our analyses, we therefore divide the sample into five different age groups (see Table 1), i.e., pre-retirement, early retirement, young old, old, and old with high longevity.

The Results section starts with a report on travel mode use among the five age groups. We then investigate and report on the relationships outlined in the theoretical model (Figure 1), combined with multi-group comparisons, in order to detect any differences between the five different segments of elderly people.

## 2. Method

### 2.1. Participants

In total, 2950 respondents (aged 60–92) from five cities in Northern Europe (Stockholm, Helsinki, Oslo, Copenhagen, Bergen) completed self-report questionnaires (online or by phone). Sample descriptives are specified in Table 2, broken down by age group. As can be seen, the distribution between men and women was generally even, with slightly more women in the oldest age group. As expected, the employment status shift substantially from pre-retirement to retirement, and from retirement to older age groups.

### 2.2. Procedure and Measures

The study is based on European public transport barometer data collected in 2018. Data collection was performed by a market research company, being carried out monthly using web surveys in combination with structured telephone interviews. The market research company (Norstat) complies with international quality standards (ISO 26362 and ISO 9001). Respondents are actively recruited, and the process is constantly monitored. The sociodemographic structure of the sample is compared to the general population to ensure representativeness. Personal invitations were sent out, including information about anonymity, volunteerism, and the right to drop out at any time. On completion of the survey, several types of rewards were made available, e.g., lottery tickets, cinema tickets, or charity donations.

In the questionnaire, the respondents were asked about sociodemographic factors (e.g., age, gender, and employment status) as well as to what extent they used six different travel modes (e.g., car as a driver, car as a passenger, public transport, bike, walking, and other), on a five-point Likert scale (ranging from 1 = never to 5 = daily).

Perceived accessibility was measured using the perceived accessibility scale (PAC) [31,38]. The respondents were asked about their perceived accessibility of daily travel by means of rating four statements (Table 3), on a seven-point Likert scale (ranging from 1 = strongly disagree to 7 = strongly agree).

Travel satisfaction was measured using the satisfaction with travel scale (STS) [24,39]. The STS combines cognitive evaluations with measures of two orthogonal affect dimensions assumed to retrospectively assess mood or emotional wellbeing during travel. The measured dimensions (one cognitive and two affective) are distinct and positively correlated constructs included in a latent higher-order measure of overall satisfaction with travel. The items in the STS were answered on seven-point scales, ranging from negative (−3) to positive (3) (see Table 4).

After the respondents had rated the perceived accessibility and satisfaction with travel scales, they were also asked about their life satisfaction (LS). Life satisfaction was measured using a single item: “Think about your daily life. All in all, how satisfied are you with your life on the whole?” and answered on a seven-point scale, ranging from (1) extremely dissatisfied to (7) extremely satisfied.

### 2.3. Analyses

The reported analysis is based on both analyses of variance (ANOVA) and partial least square structural equation modeling (PLS-SEM). The analyses were conducted in five steps: (1) testing mode use distribution between the different age groups by means of ANOVA, (2) testing the reliability and validity of the latent constructs, (3) testing the proposed model by means of PLS-SEM, (4) testing moderation by means of PLS multigroup analysis (PLS-MGA), and (5) testing the mean differences between the different age groups as regards perceived accessibility, travel satisfaction, and life satisfaction by means of ANOVA.

We applied a variance-based structural equation modeling technique of partial least squares (PLS-SEM). PLS-SEM was used because it relies on a nonparametric bootstrap procedure to test the path coefficients of the proposed model for their significance [40]. The analyses follow the recommended guidelines on how to report PLS-SEM [41].

### 2.4. Ethical Statement

This research does not fall under the Act on the Ethical Review of Research Involving Humans (Swedish Code of Statutes 2003:460), according to the local Research Ethics Committee at Karlstad University (dnr C2017/938). The research was conducted in accordance with approved research protocols. The procedure ensured that the participants were informed that their confidentiality would be maintained.

## 3. Results

### 3.1. Travel Mode Use

As shown in Table 5, mode use differs slightly between the age brackets, especially with respect to car and bike use, whereby those aged 75 and above (old and old with high longevity) report using their cars—both as drivers and passengers—and their bikes significantly less. For public transport use, no significant difference was observed; for walking, only very minor differences were found. In the table, the age group with the greatest use of each respective mode is marked in bold. As can be seen, those in early retirement seem to generally travel more frequently than the other age groups.

### 3.2. PLS-SEM

To assess the appropriateness of the measurement model, and the latent constructs (PAC and STS), convergent validity, reliability, and discriminant validity were tested. These findings are presented below, followed by measures of model fit, explained variance, and predictive relevance. Next, the test of the structural model is reported on, and the significant direct paths are visualized (Figure 2), accompanied by detailed findings regarding all direct and indirect paths (Table 6).

#### 3.2.1. Reliability and Validity of the Latent Constructs

The heterotrait–monotrait ratio of correlations (HTMT) was first assessed to confirm discriminant validity between the two latent variables. The findings were satisfactory, with the HTMT being above the suggested critical value of <0.85 [42]. Convergent validity was then evaluated using average variance extracted (AVE). AVE was above 0.50 [43], confirming a sufficient level of convergent validity. The reliability of the constructs was then assessed according to composite reliability (CR) and was found to exceed the recommended value of 0.708 [44] for both PAC and STS, providing strong support for construct reliability. In addition, the variance inflation factors (VIFs) were all below 5.0, indicating a lack of multi-collinearity between the constructs.

Predictive relevance, indicated by the Q2 measure, is a measure of how well PLS-SEM predicts the data points of the indicators. A value larger than zero indicates that the PLS path model has predictive relevance to a specific construct [40]. The Q2 value was found to exceed zero, for both STS and LS. The explained variance (R2adj) was 5% for STS and 18% for LS, and the model fit indices were acceptable (SRMR = 0.06, suggested critical value: <0.08; rms Theta = 0.15, suggested critical value: <0.12) [45]. Model fit indices should, however, be interpreted with caution in PLS-SEM as traditional and accepted threshold values may not be perfectly suited [40]. To sum up, the proposed model has acceptable validity, reliability, and predictive relevance.

#### 3.2.2. Testing the Structural Model

Figure 2 shows the significant paths of the tested overall model. All direct effects related to STS and LS are reported in Table 6. As can be seen, a significant direct effect from PAC to STS (β = 0.22) was observed, showing that higher scores in PAC relate to higher STS. The same pattern was observed between STS and LS (β = 0.36), and between PAC and LS (β = 0.15), confirming the hypothesized positive relationship between travel satisfaction and life satisfaction, and between perceived accessibility and life satisfaction. Although somewhat weaker, an indirect effect of PAC on LS, through STS (β = 0.08), was also observed.

A number of significant paths from mode use and gender to STS and PAC were observed. Specifically, being female is related to higher perceived accessibility. Using the car as a driver and walking more frequently are similarly positive as regards PAC. For satisfaction with travel (STS), a positive path was observed as regards frequent walking, while driving more frequently was related to a lower STS.

#### 3.2.3. Testing Moderation by Age Group

Partial least square multigroup analysis (PLS-MGA) was performed to test the effects of moderation by age group on the structural model [40]. Six significant moderation effects were observed, displayed in Figure 3, with straight lines corresponding to the proposed moderation in the theoretical model, and dotted lines to the additional moderation observed in the empirical material.

In line with the theoretical model (Figure 1), significant (*p* < 0.05) moderation by age group was observed in the relationships between PAC, STS, and LS. For pre-retirement, the path between STS and LS was significantly weaker than for early retirement and young old. As can be seen in Table 6, although not achieving significance in the MGA, the path was also stronger for the two oldest age groups, indicating that retirement (65 and above) is related to the increased importance to life satisfaction of satisfaction with travel.

Moderation of age group, for the relationship between PAC and STS, yielded a significantly weaker path for old with higher longevity than for pre-retirement, and marginally weaker than young old and old. These findings indicate that PAC is more strongly related to STS in the oldest age group. Taking a closer look at the path coefficients (Table 6), it can be seen that these coefficients are clearly stronger in the oldest age group than in any other age group.

Moderation of age group for the path between PAC and LS specifically shows weaker direct paths in the early-retirement (*p* < 0.05) and old with high longevity (*p* < 0.10) groups than in the other three groups. Interestingly, significant moderation was also observed in the indirect effect of PAC on LS through STS, whereby the early-retirement and old with high longevity groups have stronger paths than the pre-retirement group.

Although not part of the theoretical model, three other significant moderation effects were observed in the empirical material: (i) The path between frequency of public transport use and PAC was negative in the two youngest age groups but positive in the three elderly age groups, indicating that more frequent public transport use relates to greater PAC in some (young old, old and old with high longevity) but the opposite in others (pre-retirement and early retirement). (ii) The path between frequency of car use as a driver and STS differed between the two oldest age groups and the pre-retirement group, where the two oldest age groups had a negative path coefficient. This indicates that frequent driving is not a positive thing for the travel satisfaction of the oldest. (iii) Significant gender moderation was observed whereby being a female member of the old with high longevity group relates to a negative relationship with PAC, while positive relationships were observed in the other age groups.

#### 3.2.4. Comparisons of Means Across the Age Brackets for PAC, STS, and LS

As a final analysis, comparisons of means (analyses of variance) regarding perceived accessibility, travel satisfaction, and life satisfaction were conducted across the age groups. The means and standard deviations across the age groups are displayed in Table 7. As can be seen, only small variations in means were found between the five age groups. Statistical analyses confirmed this since no significant findings were observed.

## 4. Discussion

The main question was if perceived accessibility is related to travel satisfaction and general life satisfaction of elderly and if there are differences regarding how they travel. As suggested by the theoretical model, perceived accessibility was found to be related to both satisfaction with travel and life satisfaction, while travel satisfaction was directly associated with life satisfaction. These findings show that the abilities of the elderly to participate in daily activities constitute an aspect that likely affects their overall life satisfaction, that subjective evaluations of travel are important to the elderly, and that both the possibility and ability to get around using the transport system relates to cognitive and emotional experiences while doing so.

Although these direct effects are important and interesting in themselves, it is equally important to recognize that the proposed moderating effects of age groups was also confirmed, and especially so with respect to the old with high longevity group (80+) when compared with other groups. As Haustein et al. [46] point out, there is not much previous knowledge of the oldest group of the elderly in research on daily travel, making this group appropriate to highlight in the following discussion. For this group, the meaning of perceived accessibility for life satisfaction diminishes in comparison to elderly people below the age of 80. Thus, for the oldest, other things than accessibility appear to be of importance to satisfaction in life. It is likely that elderly in this group no longer have the same need to reach an array of daily activities as they previously did in life, making accessibility less important to overall life satisfaction, as well as other aspects of life more related to life satisfaction, e.g., health or family. In fact, previous findings [32,35] show that individuals start to engage more in in-home activities and less in out-of-home activities as their age increases. This further corresponds to our finding that those over 80 experience the lowest levels of perceived accessibility out of all the groups, albeit with the highest levels of life satisfaction (alongside the 75–79 bracket). Thus, for the oldest, more and more activities may be perceived as being too far away or too complicated to reach; subsequently, in-home activities become more important for their overall satisfaction in life. However, when the oldest do travel, their travel needs to be of a good standard; thus, our findings show that travel satisfaction still matters for their life satisfaction. In fact, in line with previous research on the links between satisfaction with travel and life satisfaction [25], our results show that satisfaction with travel is important in all age groups. Thus, even if perceived accessibility matters less for life satisfaction in some of the elderly groups, the quality and experiences of daily travel are always important. This is evident in the moderated mediation of satisfaction with travel for the indirect relationship between perceived accessibility and life satisfaction, whereby the indirect relationship is stronger in old with high longevity than it is in other age groups.

Looking more closely at the travel behavior of all the groups, the oldest once again stand out from the rest. They generally travel less than the other age groups and are, presumably, more reliant on the quality and overall experience of their travel due to issues of ageing and health. Although they travel by car and bicycle significantly less than the elderly aged up to 74, they still manage to walk and use public transport to much the same extent as other elderly people. Interestingly, the results show that frequent walking and car use are positively associated with perceived accessibility across all the age groups, while frequent public transport use is positive for those over 70 and negative for those under 70. This indicates that members of the elderly age groups travel to destinations using public transport that they perceive to be accessible, or that using public transport per se may be a social activity of choice, which affects their accessibility. Greater use of public transport before retirement, however, is associated with lower levels of accessibility. This may be due to the fact that the majority of the elderly in this group are still working (<60%, Table 2), and that public transport may not be perceived to be accessible enough to meet the travel needs of these people.

Looking at the main effects of gender and perceived accessibility, it can be seen that women perceive their accessibility as higher than men do. This is in line with previous research on gender differences in perceived accessibility [37,38]. However, the moderation analysis shows that, for the oldest, women actually perceive their accessibility as lower than men do. Thus, the effect of gender partly contradicts previous findings. Unfortunately, we do not have an adequate explanation for this finding.

Bearing in mind that the groups in the present study are based on age (a five-year span), and not on life biographies, which are likely to extend across age segments, there is a possibility that the inclusion of other determinants in the models, e.g., health status, economic factors, relationship status, activities, or attitudes, would alter our findings somewhat. For instance, there is a possibility that, even though perceived accessibility and satisfaction with travel appear to be associated with life satisfaction, the effect of these relationships is marginal compared with other things in life, especially in the oldest group. In the future, the inclusion of further variables in the models could help to better explain some of the differences between the age groups. For instance, the oldest group may as well travel less for economic reasons, as for health issues, or be satisfied with traveling less due to the availability of social contacts nearby.

Finally, it is interesting to note that levels of perceived accessibility, satisfaction with travel, and life satisfaction do not differ significantly between any of the age groups. This may indicate that, while there are differences in how these groups travel, e.g., frequency and mode use, they generally seem to be able to travel using modes that offer them acceptable levels of travel satisfaction and accessibility. However, this finding should be set in relation to previous studies which have come to the conclusion that levels of perceived accessibility decline with age [37,47] and that subjective factors (e.g., attitudes) are better predictors of levels of perceived accessibility than sociodemographic factors are [37]. Moreover, individuals with mobility impairments caused by aging have been found to experience significantly lower levels of accessibility than other impairment groups [48]. As the majority of the elderly in all the age groups are still reliant on their private cars for daily travel (with 60%–70% using their cars as drivers or passengers either daily or a few times a week, in line with other findings from Northern Europe [49]), their levels of perceived accessibility will likely be affected if they are forced to use more sustainable modes. An enforced change from the car to sustainable travel modes was found, in a recent hypothetical study [50], to have a negative effect on perceived accessibility among frequent and less frequent car users. Given our findings, where frequent public transport use was negatively associated with perceived accessibility by the two youngest groups of elderly (a fraction of whom are still working), but positively associated by the old with high longevity, we recommend further research in order to obtain conclusive answers as regards how changes in transport mode use affect the different cohorts of the elderly.

## 5. Conclusions

This study offers original insights into the relationship between perceived accessibility, satisfaction with travel, and life satisfaction in different segments of the elderly. Through an empirical study of the travel behavior of elderly people of different ages, we show that perceived accessibility is significant to overall life satisfaction. For some of the elderly, this relationship is somewhat weaker; however, the concepts are still indirectly related via satisfaction with travel. We further show that satisfaction with travel is always important to older people’s life satisfaction, although slightly less so for the youngest old. We are thus able to conclude that the moderating role of age among the elderly clearly indicates that we should not treat the elderly as one homogeneous group in transport planning. Based on our findings, we conclude that a continuous and long-term quality work is important with a special focus on segments of older groups in order to improve relevant and significant attributes. Grounded on previous research [31], we also recommend a special focus on safety, although additional knowledge is called for on the relationship between feeling safe, perceived service quality, and perceived accessibility. To gain a better understanding of the role of travel in old people’s lives, we encourage future studies to include additional potential factors (e.g., attitudes and feelings of safety) of influence in the model.

## Figures and Tables

**Figure 1 ijerph-16-04498-f001:**
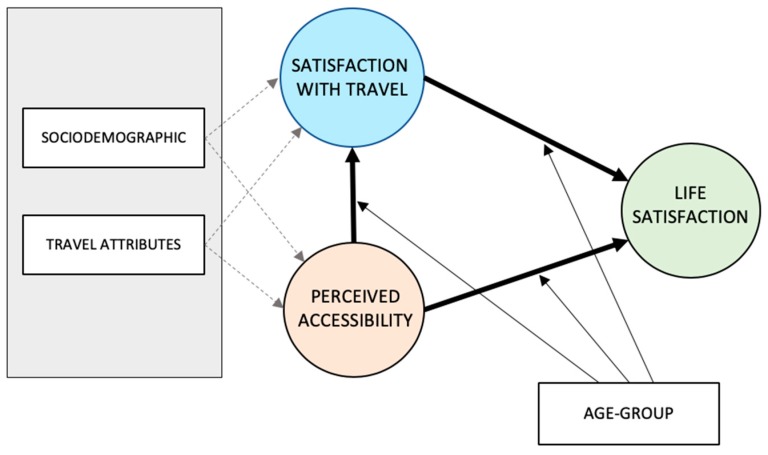
Theoretical model.

**Figure 2 ijerph-16-04498-f002:**
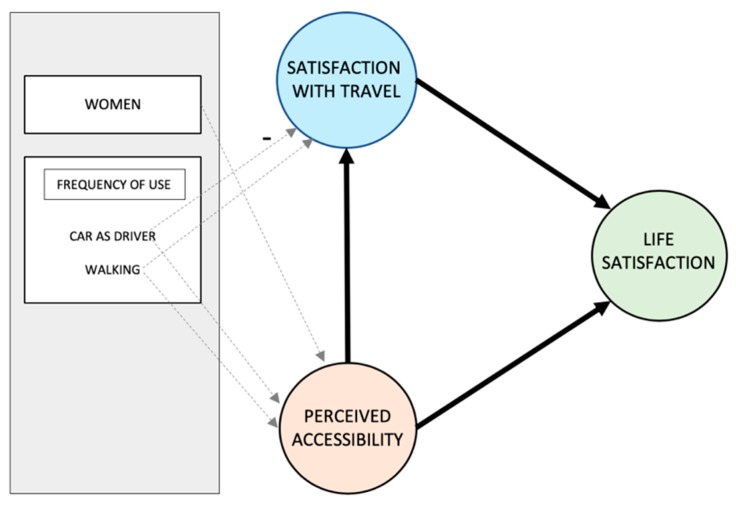
Significant (*p* < 0.05) direct effects on perceived accessibility, travel satisfaction and life satisfaction. Note: The path between car driver and STS is marked with a minus sign, indicating a negative relationship. The remaining paths are positive.

**Figure 3 ijerph-16-04498-f003:**
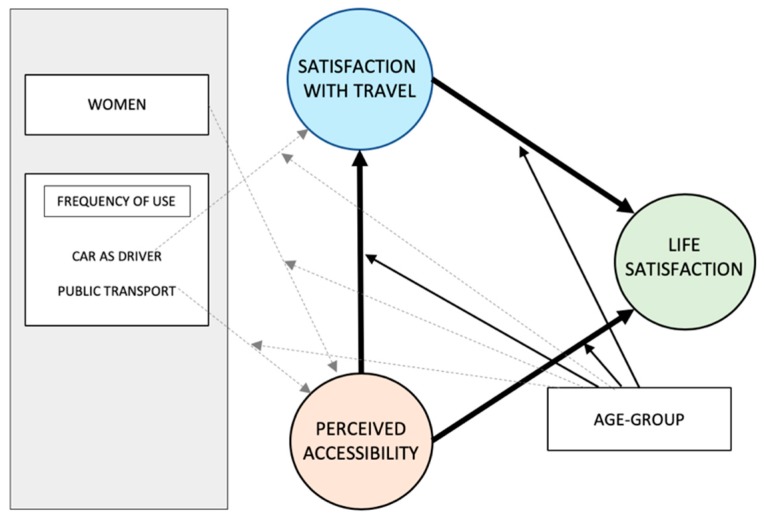
Significant (*p* < 0.05) moderating effects of age group.

**Table 1 ijerph-16-04498-t001:** Five age groups in the older population.

Age Groups	Year of Birth	Age in 2019
Pre-retirement	1958–1954	60–64
Early retirement	1953–1949	65–69
Young old	1948–1944	70–74
Old	1943–1939	75–79
Old with high longevity	1938–	80–

**Table 2 ijerph-16-04498-t002:** Sample descriptives broken down by age group.

Descriptive		Age Group				
		Pre-Retirement	Early Retirement	Young Old	Old	Old with High Longevity
Age		(60–64)	(65–69)	(70–74)	(75–79)	(80–)
Number of participants (*N*)		866	677	707	363	337
Women (%)		45.5	53.6	48.1	50.4	58.2
Employment status (%)						
	Working full time	51.2	10.0	1.7	0.3	-
	Working part time	10.5	5.2	3.5	0.8	2.4
	Retired	31.9	83.3	93.4	98.3	98.3
	Other not specified	6.2	1.3	1.4	0.6	0.6

**Table 3 ijerph-16-04498-t003:** Statements included in the perceived accessibility scale (PAC).

PAC Item	Wording
1	Considering how I travel today, it’s easy to do (daily) activities.
2	Considering how I travel today, I’m able to live my life as I want to.
3	Considering how I travel today, I’m able to do all the activities I prefer to do.
4	Considering how I travel today, access to my preferred activities is satisfying.

**Table 4 ijerph-16-04498-t004:** Statements included in the satisfaction with travel scale (STS).

Cognitive Quality Evaluation								
	“Think about your daily travel. How do you experience your daily travel in general? My trips…”							
work poorly, are of a low standard, the worst imaginable.	−3	−2	−1	0	1	2	3	work well, are of a high standard, the best imaginable.
**Affective Evaluation of Feelings of Boredom vs. Enthusiasm**								
	“Think about your daily travel. How do you experience your daily travel in general? I feel …”							
very bored, tired, fed-up.	−3	−2	−1	0	1	2	3	very enthusiastic, alert, engaged.
**Affective Evaluation of Feelings of Stress vs. Relaxation**								
	“Think about your daily travel. How do you experience your daily travel in general? I feel …”							
very stressed, worried, hurried.	−3	−2	−1	0	1	2	3	very relaxed, calm, confident.

**Table 5 ijerph-16-04498-t005:** Use of different modes (daily or a few times a week, in %) and group comparisons between age groups, by Kruskal–Wallis ANOVAs.

Mode	Pre-Retirement	Early Retirement	Young Old	Old	Old with High Longevity	Group Comparisons Using Kruskal–Wallis ANOVAs (H)
Car as driver	56.0	56.7	53.3	46.8	36.9	H = 60.08, *p* < 0.001
Car as passenger	19.0	22.3	20.1	13.6	13.4	H = 26.98, *p* < 0.001
Public transport	35.5	37.4	33.0	34.7	37.0	H = 2.73, *p* = 0.601
Bicycle	22.9	22.8	20.1	14.8	11.8	H = 107.63, *p* < 0.001
Walking	29.1	35.2	31.9	28.0	29.8	H = 21.86, *p* < 0.001
Other *	3.4	1.1	1.4	1.7	2.8	

* All other reported modes were subsumed in the category “other”, e.g., taxi, motorcycle, boat.

**Table 6 ijerph-16-04498-t006:** Direct effects reported in PLS-SEM of mode use, gender, perceived accessibility (PAC), satisfaction with travel (STS), and life satisfaction (LS).

Direct Effects	β	t	*p*
Bicycle → PAC	0.017	0.979	0.328
Bicycle → STS	0.010	0.517	0.605
Car as driver → PAC	0.146	7.152	<0.001
Car as driver → STS	−0.048	2.538	0.011
Car as passenger → PAC	−0.021	1.046	0.296
Car as passenger → STS	−0.022	1.119	0.263
PAC → LS	0.148	6.395	<0.001
PAC → STS	0.219	11.564	<0.001
Public transport → PAC	−0.024	1.067	0.286
Public transport → STS	−0.014	0.706	0.480
STS → LS	0.359	15.006	<0.001
Walking → PAC	0.093	4.708	<0.001
Walking → STS	0.062	3.153	0.002
Women → PAC	0.045	2.316	0.021
Women → STS	−0.031	1.570	0.117

**Table 7 ijerph-16-04498-t007:** Means and standard deviations (in brackets) across the age groups for PAC, STS, and LS.

Construct	Age Group				
	Pre-Retirement	Early Retirement	Young Old	Old	Old with High Longevity
Perceived Accessibility	5.64(1.2)	**5.81**(1.2)	5.77(1.2)	5.74(1.2)	5.58(1.3)
Satisfaction with Travel (1–7)	4.56(1.5)	4.81(1.5)	4.92(1.4)	**4.97**(1.4)	4.79(1.4)
Life Satisfaction (1–7)	5.56(1.4)	5.74(1.3)	5.68(1.3)	**5.80**(1.2)	**5.80**(1.1)

Note: The age group with the highest mean for each of the constructs is marked in bold.

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
