# Peer review of "Perceived Accessibility, Satisfaction with Daily Travel, and Life Satisfaction among the Elderly"

_ijerph, 2019, doi:10.3390/ijerph16224498_

Round 1
Reviewer 1 Report
This paper discusses an empirical investigation into the relationship between perceived accessibility, satisfaction with travel, and life satisfaction for different segments of elderly respondents. The paper is well researched, organized and written. It would be useful to discuss how representative the survey sample is of the population in the five cities covered in the study. In addition, it would be relevant if the paper offered a few policy implications of its empirical findings. Finally, lines 406-466 are exact copies of lines 345-405 and need to be removed.
Author Response
This paper discusses an empirical investigation into the relationship between perceived accessibility, satisfaction with travel, and life satisfaction for different segments of elderly respondents. The paper is well researched, organized and written.
It would be useful to discuss how representative the survey sample is of the population in the five cities covered in the study.
As stated in section 2.2, we base our study on European public transport barometer data collected in 2018. Data collection was performed by a market research company, being carried out monthly using web surveys in combination with structured telephone interviews. The market research company (Norstat) complies with international quality standards (ISO 26362 and ISO 9001). On page 6, we have added the following sentence:
Respondents are actively recruited and the process are constantly monitored. The sociodemographic structure of the sample is compared to the general population to ensure representativeness.
In addition, it would be relevant if the paper offered a few policy implications of its empirical findings.
As a response to you and Reviewer 2 we have included additional text on page 12 (section 5 “Conclusions”).
Based on our findings, we conclude that a continuous and long-term quality work is important with a special focus on segments of older groups in order to improve relevant and significant attributes. Grounded on previous research [31], we also recommend a special focus on safety although additional knowledge is called for on the relationship between feeling safety, perceived service quality, and perceived accessibility.
Finally, lines 406-466 are exact copies of lines 345-405 and need to be removed.
Thank you for noticing this doubling. The text has been removed.
Reviewer 2 Report
This is a reliable article with scientific research methods and analysis. Through an empirical study of travel behaviors of elderly people of different ages, this paper presents the relationship between perceived accessibility, satisfaction with travel and life satisfaction in different segments of the elderly and points out that the elderly should not be treated as a homogenous group. I would like to suggest the authors highlight the importance of the study in Conclusions by giving examples of its use in transport planning or other research, and listing some additional potential factors as it said in the last sentence.
Author Response
This is a reliable article with scientific research methods and analysis. Through an empirical study of travel behaviors of elderly people of different ages, this paper presents the relationship between perceived accessibility, satisfaction with travel and life satisfaction in different segments of the elderly and points out that the elderly should not be treated as a homogenous group.
I would like to suggest the authors highlight the importance of the study in Conclusions by giving examples of its use in transport planning or other research, and listing some additional potential factors as it said in the last sentence.
Thank you for this comment. We have added additional text accordingly on page 12 (section 5 “Conclusions”). We have also listed some additional potential factors (e.g., attitudes and feelings of safety) in the last sentence.
Based on our findings, we conclude that a continuous and long-term quality work is important with a special focus on segments of older groups in order to improve relevant and significant attributes. Grounded on previous research [31], we also recommend a special focus on safety although additional knowledge is called for on the relationship between feeling safety, perceived service quality, and perceived accessibility.